# Multiscale Numerical Modeling for Prediction of Piezoresistive Effect for Polymer Composites with a Highly Segregated Structure

**DOI:** 10.3390/nano11010162

**Published:** 2021-01-10

**Authors:** Oleg V. Lebedev, Alexander N. Ozerin, Sergey G. Abaimov

**Affiliations:** 1Center for Design, Manufacturing and Materials, Skolkovo Institute of Science and Technology, 143026 Moscow, Russia; s.abaimov@skoltech.ru; 2Moscow Institute of Physics and Technology, Institutsky Lane 9, Dolgoprudny, 141700 Moscow, Russia; 3N.S. Enikolopov Institute of Synthetic Polymer Materials of RAS, Profsoyuznaya St. 70, 117393 Moscow, Russia; ozerin@ispm.ru

**Keywords:** polymer composites, carbon nanoparticles, piezoresistive effect, electrical conductivity, ultrahigh-molecular-weight polyethylene

## Abstract

In this work, the piezoresistive effect for a polymer nanocomposite with a highly segregated distribution of conductive filler was investigated. As a base polymer for the investigated nanocomposites, ultrahigh-molecular-weight polyethylene, processed in a solid state (below melting point), was used. Multiwalled carbon nanotubes (MWCNTs) were used as a nanofiller forming a highly segregated structure in between polymer particles. A numerical multiscale approach based on the finite element method was proposed to predict changes in the conductive structure composed of MWCNTs in response to uniaxial deformation of the material. At the nanoscale, numerical simulations were conducted for uniformly distributed MWCNTs providing confinement of the filler to a two-dimensional layer with a high volume fraction of the filler in between two polymer particles. At the microscale, the piezoresistive response to uniaxial deformation for the three-dimensional highly segregated structure reconstructed from experimental data was investigated numerically. The embedded element method was implemented to conduct a realistic and computationally efficient simulation of MWCNT behavior during deformation of the nanocomposite. The results of numerical simulations were compared with the experimental data to prove the correctness of assumptions used in the modeling.

## 1. Introduction

In recent years, the ability to nondestructively control the deformation of a material by correlating its characteristics with its deformational state has become more and more in demand [1]. Measuring electrical conductance of the material in the course of its deformation is one of the efficient ways of doing this [2]. Most polymer materials are not characterized by a significant level of electrical conductivity, unless they are modified by filling them with an electroconductive filler [3]. The type, concentration, and distribution of the filler are the parameters that determine the electrical conductivity of the polymer. By varying these parameters, it is possible to widen the area of the nanocomposite applicability, starting with electrostatic protection and electromagnetic interference shielding and finishing by using nanocomposites as metal replacements. Taking into account the sensitivity of the percolation network to deformation, it is also possible to apply such nanocomposites as sensors [4].

Multiwalled carbon nanotubes (MWCNTs) are widely used due to their high electrical conductivity and aspect ratio [5,6,7]. According to percolation theories, high values of a filler’s aspect ratio provide low values of the percolation threshold. This is important for practical applications, not only in terms of cost considerations, but also because a minimal negative impact of the filling on the mechanical characteristics of the polymer material is desirable [8,9].

Along with the filler characteristics, its distribution and dispersion in a matrix can also significantly influence properties of a polymer nanocomposite, especially its electrical conductivity [10,11]. The traditional way to achieve electroconductivity of a polymer is to uniformly distributee the conductive nanofiller. Nanocomposite systems, in which the filler distribution and dispersion are indeed close to uniform, are well described experimentally and analytically, and they are often characterized by predictable mechanical and electroconductive properties [12,13,14]. However, uniform dispersion of nanosized filler is difficult to achieve in practice due to the tendency of nanoparticles to agglomerate [15,16,17,18]. The agglomeration of nanoparticles results in an increase in the percolation threshold and filler volume fraction, required to achieve a desirable level of electrical conductivity [8,15]. Furthermore, when the electrical conductivity of MWCNT nanocomposites is modeled, analytically or numerically, agglomeration of MWCNTs often results in wrong estimation of model parameters, which are usually optimized to fit the experimental data. 

Segregation of nanoparticles presents a different way to achieve high levels of electrical conductivity, which is not prone to these difficulties. As demonstrated earlier, the formation of an interconnected structure with densely packed filler particles (compared to the particle-free regions in the rest of the composite volume)—the so-called segregated structure—is a good way of significantly reducing the percolation threshold value for the resulting nanocomposite [11,19], providing, of course, the interconnectivity of the densely populated regions. One way to create such a structure is to mix nanofiller with polymer powder, where, after processing, the layers of nanoparticles form a foam-like structure, enveloping the polymer granules. In this case, the decrease in the percolation threshold in comparison to the case of the uniform filler dispersion is guaranteed by two factors: first, polymer particles are filler-free inside, which lowers the required fraction of the nanofiller and, second, the formation of the structure surrounding the polymer particles self-generates the interconnectivity of the regions densely filled with nanoparticles.

One of the polymers that received substantial attention in the last few decades is ultrahigh-molecular-weight polyethylene (UHMWPE), which is often used as a base polymer for nanocomposites with segregated structures made of nanosized filler [20,21]. Due to the high viscosity of the UHMWPE melt, a limited number of processing methods are available for this type of polymer. One of the most common methods of obtaining nanocomposites based on UHMWPE is hot (above the melting point of PE) sintering of a mechanical mixture of UHMWPE powder and nanofiller, which leads to formation of a segregated structure [21]. In this case, due to the processing above the melting point, partial dispersion of nanofiller in the matrix is present.

A special type of UHMWPE reactor powder can be processed in a solid state (below the melting point of PE) and, subsequently, oriented to high degrees of orientation [22] (Figure 1a). It was demonstrated that the presence of a filler even in a very high amount does not significantly affect the process of orientation, making it possible to obtain strong and highly flexible electroconductive materials. Nanocomposites based on this special type of UHMWPE and processed in a solid state via compression molding are characterized by an extreme segregation of the filler and by very low values of the percolation threshold [20,23] in comparison to the hot-molded nanocomposites. In the case of solid-state processing of polymer and MWCNT powders, since polymer melting is not involved, MWCNTs only cover polymer powder particles, mostly not penetrating their volume. As such, the dispersion of nanofiller in the matrix is not actually present, and we can talk only about the three-dimensional distribution of extremely segregated nanostructure in the volume of the nanocomposite (Figure 1d) and about the distribution of the MWCNTs on the surfaces of polymer powder particles (Figure 1c,d). Therefore, we reserve the term “highly segregated structure” for the nanocomposite manufactured via solid-state processing to distinguish the resulting nanostructure from the melt processing case. In our study, we investigate only highly segregated structures as the result of solid-state processing.

The easiest way to establish a relationship between the electrophysical properties of a material and its deformational state is by using the knowledge on the initial distribution of the filler particles, followed by the analysis of changes in mutual position and contacts of particles during deformation of the material. For nanocomposites filled with MWCNTs, it is also necessary to follow the changes in the individual orientations of the MWCNTs [24,25,26]. Many research groups made attempts to numerically predict the electrophysical properties of composites filled with CNTs [25,27,28], but there is still a need for a working numerical model that takes into account realistic filler behavior during deformation, i.e., considering possible complex conformations of MWCNTs compacted to a thin layer in between two polymer particles.

Most numerical models that currently exist are based on finite element methods (FEM) [1,29,30,31,32,33,34]. These models allow estimating at different degrees of accuracy the mechanical characteristics and electroconductivity of nanocomposites filled with particles of different types. However, considering that a high volume fraction of filler is required for percolation, conductivity and, especially, piezoresistivity calculations for systems with filler content above the percolation threshold can be very computationally demanding, especially in three dimensions (3D) [29,32,33]. Thus, these approaches do not deal with systems containing a large number of particles, while also making serious simplifications to afford reasonable computational times. 

To increase the computational efficiency of FEM simulations, the embedded element method for mesh creation can be used [32,35,36,37]. This method implies the creation of two separate independent meshes—for the matrix and for the filler—without the necessity to generate an identical node network at the interface. Utilization of the embedded elements for the filler reduces the total number of finite elements by more than on order of magnitude, allowing the modeling of regions densely filled with filler particles.

At the previous stage of this work, the piezoresistive response of a thin MWCNT layer deposited on a polymer surface was investigated experimentally and numerically [36]. A good correspondence between numerical and experimental results was observed. A study of the results’ sensitivity to the model’s input parameters was performed. It was determined that, in the investigated range of values, the conductance response was not very sensitive to the thickness of the thin layer, as well as to the changes in the volume fraction of MWCNTs. At the same time, the statistical length of MWCNTs was proven to be the critical parameter, greatly influencing the results of numerical simulations. 

The next stage of the work is to apply the knowledge obtained at the nanoscale for the thin two-dimensional MWCNT layer to construct the three-dimensional segregated structure at the microscale, whose homogenization will provide the piezoresistive response of the nanocomposite at the macroscale, thereby involving multiscale modeling. We develop a computationally efficient numerical approach capable of predicting the electrical conductance response to uniaxial deformation for a nanocomposite with highly segregated structure manufactured via solid-state processing of UHMWPE. 

## 2. Materials and Methods

### 2.1. Materials

UHMWPE nascent reactor powder with a special nodular morphological structure [22] (St. Petersburg Branch of Institute of Catalysis, St. Petersburg, Russia) was used as a matrix for the nanocomposites (molecular weight 5 × 10^6^, bulk density of the reactor powder 0.058 g/cm^3^). The morphology of the used UHMWPE powder is shown in Figure 1a.

As the filler, NC 7000 type MWCNTs (Nanocyl, Sambreville, Belgium) were used in this work [38]. The specification of the MWCNTs is presented in Table 1. The morphology of the filler is shown in Figure 1b. The choice of NC 7000 MWCNTs as a filler was due to the narrow distribution of their diameter and size, as well as their high purity and affordable price. The MWCNTs of choice are also characterized by a low degree of entanglement, which is a significant issue for commercially available single-walled carbon nanotubes (SWCNTs), some of which are known to be grown in bundles [39].

### 2.2. Methods

#### 2.2.1. Nanocomposite Processing

Samples of UHMWPE-based nanocomposites with a segregated structure made of MWCNT filler were manufactured using a multi-step procedure. First, the components (98 wt.% UHMWPE + 2 wt.% MWCNT) were mixed in a volatile nonsolvent liquid (3 g of the UHMWPE + 2 wt.% MWCNT mixture added to ~50 mL of hexane) using a CUD-500 ultrasonic disperser from Criamid, Moscow, Russia (5–10 min, 250 W, 30 kHz). Next, the UHMWPE particles covered with MWCNTs were deposited on a filtering paper to remove the excessive liquid and were left to dry for a day at room temperature. The uniformity of the distribution of nanoparticles on the surface of the granules of the UHMWPE reactor powder was evaluated using electron microscopy. The results of microscopy (Figure 1c) showed, among the surfaces of the UHMWPE reactor powder densely populated with MWCNTs, the presence of areas not covered by nanoparticles (probably, as a result of the low 2 wt.% utilized). Subsequent compression of this mixture at room temperature (20 °C) in a closed mold under pressure of 200 MPa produced the nanocomposite plates with dimensions 50 mm × 12 mm × 0.5 mm. The distribution of nanoparticles in the compressed samples was also studied using electron microscopy. Formation of a highly segregated structure was assured by the nature of the solid-state processing below the melting point of the polymer with no dispersion of the nanofiller in the matrix. This is confirmed by scanning electron microscopy results in Figure 1d, where the fracture surface of the manufactured nanocomposite demonstrates the surface of a polymer powder particle still covered by MWCNTs after compression molding. It is noteworthy that the microscopy observed a denser MWCNT coating on the surface of UHMWPE particles after compression (Figure 1d) in comparison to the distribution of MWCNTs before compression (Figure 1c), which was attributed to the reduction in volume of UHMWPE particles after compression (~20-fold). The electrical conductivity of the formed nanocomposite plates was measured using the four-probe resistance measurement method and an Agilent 34401A multimeter.

#### 2.2.2. Methods of Investigation

The electron microscopy studies were performed using electron microscope Supra 50 VP LEO (Carl Zeiss AG, Oberkochen, Germany) with INCA Energy + Oxford microanalysis system and scanning electron microscope SU8000 (Hitachi, Ibaraki, Japan). The fracture surfaces were prepared via freeze-fracturing of the test samples at the liquid nitrogen temperature.

To study the deformational behavior of the electrical conductivity of nanocomposites, test complex 5969 (Instron, Norwood, MA, USA) was used in tension mode with a nominal strain rate of 0.1 min^−1^. The base of the samples was 10 mm. Samples were wrapped in thin copper foil pieces at the both ends, which, in turn, were insulated from the clamps of the testing machine with fine-grained sandpaper. During the deformation, a voltage of 2 V was supplied to the foil by power source AKIP 1147/1 (Prist, Moscow, Russia), while using an analog-to-digital converter with a frequency of 100 Hz. The value of the electric current, passing through the sample and measured by picoammeter 6485E (Keithley, Cleveland, OH, USA), was recorded. The obtained values of the electric current were converted into the values of the relative electrical conductivity—the ratio of the value of electrical conductivity in the deformed state to its value in the undeformed sample.

The compacted UHMWPE + 2 wt.% MWCNT nanocomposite samples were studied layer-by-layer using electron microscope Helios NanoLab 660 (Thermo Fisher Scientific, Waltham, MA, USA) and focused ion beam etching with a layer spacing of 100 nm and a photographic region of 20 μm × 25 μm (Figure 2a,b). The total number of images (layers) was 290. Figure 2c presents one such slice of the nanocomposite as an example. Areas densely populated with MWCNTs can be clearly observed; however, the trustworthy reconstruction of the cross-section of the interconnected segregated structure is impossible using the image processing of just one slice. To overcome this difficulty, special image processing was implemented to process images not separately, but in sequence, confirming the location of the segregated structure by its presence around a polymer granule through a sequence of several cuts and ignoring uncorrelated noise between images. The result of such a reconstruction is presented using dashed lines in Figure 2d for a slice.

#### 2.2.3. Numerical Modeling

The first stage of the numerical modeling consisted of studying the deformational behavior of a thin layer of densely packed MWCNTs in between two polymer particles. In mechanical simulations, MWCNTs were modeled by chains of rigid segments, similar to statistical (Kuhn) segments used for polymer chain representation, which can freely rotate around the joint points. This assumption is based on studies [40] demonstrating that a CNT can deform freely (without application of external loading) at lengths exceeding the value of its statistical length. Efficiency of a similar representation by segments was demonstrated earlier in percolation threshold calculations for wavy nanotubes [28,41]. In our study, MWCNTs were represented as chains of two-node 3D truss segments. These types of elements are characterized by the ability to rotate freely around the joints between the elements. The number of elements (segments) in a single MWCNT was determined by dividing the MWCNT length (1.5 μm in this work) by the equivalent of the statistical (Kuhn) length for a nanotube, the value of which (~250–500 nm) was estimated using literature data [40] and the results of electron microscopy. Specifically, the value of statistical length of 375 nm was used. 

A special type of volume element was constructed to imitate periodical boundary conditions for MWCNT network conductance calculations (Figure 3). First, the initial points (“seeds”) of the MWCNTs were distributed uniformly in a volume of 0.1 μm × 5 μm × 5 μm. The number of MWCNTs was chosen according to the volume fraction of the filler in the conductive layer (55%), obtained experimentally at the previous stage of study [36]. The layer thickness of 100 nm was calculated as the thickness of a conductive layer enveloping UHMWPE grains in solid-state processed UHMWPE/MWCNT nanocomposites with ~1% total volume fraction of the filler in the matrix.

From each “seed”, an MWCNT was “grown” as follows: at the distance from the initial point corresponding to the statistical length, the next point was created at a random angle, thus forming the first segment of the MWCNT. A finite element (FE) was created as spanning this distance. The process was repeated with the previously created point serving as a new reference one.

The coordinates of the new points were limited by the dimensions of the selected volume, forcing the new MWCNT segments to be “deflected” inward at the boundaries of the MWCNT layer. While in the layer thickness direction, this confinement is natural for the two-dimensional structure of the layer in between two polymer particles, whereas the in-plane confinement of MWCNTs is artificial and caused by the finite size of the modeled layer area. To avoid consequences of this effect, only the “inner” part (0.1 μm × 2 μm × 2 μm) of the populated layer volume was studied (surrounded by dashed line in Figure 3a). MWCNTs that did not have any points inside the “inner” volume were removed. This allowed us to imitate periodic boundary conditions for conductance of the “inner” volume as it is shown in Figure 3a. Electrical conductance of the “inner” volume was calculated at the next step. 

The discussed model technique was selected for two reasons. First, the reason for not using the representative volume element as obtained by truncation of the initial volume to the boundaries of the “inner” volume was because, in that case, some MWCNT segments cut by the new boundaries might have lengths shorter than the statistical length. The presence of segments of different lengths can result in unpredictable outcomes [36] and, therefore, is undesirable. Second, periodic boundary geometry could not be created for the truncated volume due to the specifics of the meshing method selected in this work—the embedded element method—that eliminates all degrees of freedom of MWCNT elements. Discussion on the advantages and disadvantages of the embedded element method applied to similar systems can be found in our publications [35,36].

The resulting distribution of MWCNT segments was imported into SIMULIA Abaqus^TM^ FEA software suite. A two-node 3D truss element (T3D2) mesh was created for the MWCNT structure, with the element cross-section parameters estimated from the MWCNT manufacturer data (Table 1). The following properties were assumed for matrix and MWCNT materials, respectively: elastic modulus of 100 MPa and 1000 GPa (data taken from the NC7000 MWCNT specifications) and Poisson coefficient of 0.46 and 0.3 [42]. Since the contrast between the mechanical properties of the matrix and MWCNTs was very high, the precise value of the matrix elastic modulus was considered to be irrelevant.

The matrix mesh was generated using equally sized eight-node cubic elements (C3D8, edge size 20 nm) for the volume surrounding MWCNTs (Figure 3). Next, the MWCNT mesh was embedded into the matrix mesh.

As an example of numerical modeling, we consider in this section the tensile displacement applied in plane, to the planes coplanar with the upper and lower faces of the inner volume, denoted by the upper and lower dashed lines in Figure 3a. The changes in coordinates of all MWCNT nodes were recorded over the course of the volume deformation. The obtained dataset of coordinates of MWCNT nodes for different degrees of deformation was exported for the conductance calculations.

In order to calculate the conductance of the investigated volume element at a given deformation, the intrinsic resistance of MWCNTs, provided by the ballistic transport of electrons along them, was assumed to be negligible compared to the tunneling resistance of joints in between MWCNTs or between an MWCNT and an electrode [43,44]. The relative conductance (the ratio of the conductance for a selected degree of deformation to the conductance of nondeformed volume) was obtained using Kirchhoff’s circuit laws for the grid of unit contact resistances and unit input and output electrical currents flowing through the upper and lower faces of the “inner” volume (upper and lower dashed lines in Figure 3a). The efficiency of a similar approach for conductance calculations in CNTs networks was demonstrated earlier in several papers [25,27,34,41]. All distances between the MWCNT elements equal to or less than two MWCNT radii were considered as contacts. To take into account the tunneling effect, the law of the exponential decrease in contact conductance was implemented with a characteristic distance of 0.5 nm [43,44]. The contact between electrodes and an MWCNT is formed if this MWCNT crosses one of the faces of the “inner” volume to which the deformation is applied (upper or lower dashed lines in Figure 3a). One of the MWCNTs (chosen at random) is characterized by a zero value of electrical potential. Next, all other MWCNTs potentials are calculated using Kirchhoff’s law.
(1)$$I_{i}=\;Q_{ij}U_{j},$$
where *I_i_* is a vector of differences between input and output electrical currents for all MWCNTs and electrodes, *U_j_* is a vector of electrical potentials (voltages) for all MWCNTs and electrodes, and *Q_ij_* is a matrix of conductance values for all possible MWCNT/MWCNT joints and for MWCNT/electrode junctions with respect to the direction of the current flow. The effective conductance of the “inner” volume was calculated as the reciprocal voltage difference between the first electrode and the second.

## 3. Results and Discussion

The electrical conductivity of nanocomposite samples with various concentrations of MWCNTs was measured, and the percolation threshold was estimated. In the result, the percolation threshold was several times lower than that predicted by the classical percolation theory (~0.1 vol.% vs. > 0.4 vol.%), which indicates the formation of a segregated structure of the filler. For the subsequent tests, samples were manufactured with the concentration of MWCNTs known to be higher than the percolation threshold.

The piezoresistivity of nanocomposites was investigated for a concentration of MWCNTs in the range 1–5 wt.%. For each concentration, three samples were tested, and the results of the tests were averaged. All resulting dependencies of the relative conductances (conductance in the deformed state divided by the conductance in the undeformed state) on the elongation were close to linear, with the slopes of the fitted linear function being within the scatter around the slope determined for UHMWPE + 2 wt.% MWCNT nanocomposites ((−7.0 ± 0.2) × 10^−3^). Therefore, it was decided to use only UHMWPE + 2 wt.% MWCNT nanocomposites for experimental verification of the numerical model.

A series of samples of nanocomposites containing MWCNTs in various concentrations was also subjected to repeated pressing in a heated mold under a pressure of 200 MPa and a mold temperature of 160 °C for 5 min. The samples thus obtained had two orders of magnitude lower conductivity compared to the nanocomposites based also on UHMWPE, but obtained without the stage of processing through the melt. This effect was attributed to the fact that, although UHMWPE does not have the ability to flow in a melted state due to its extremely high viscosity, there is possibility for MWCNT particles to penetrate into the polymer particle to a certain depth, causing partial dispersion. It is also possible for the polymer melt to get into the contact zone between the nanoparticles. This leads to a decrease in the density of the layer of nanoparticles and the number of contacts between them, which, in turn, leads to a decrease in the electrical conductivity of the nanocomposite sample as a whole.

The images obtained via focused ion beam etching electron microscopy were processed in the ImageJ software package, followed by the processing of images in a sequence, to obtain a three-dimensional surface of the highly segregated structure reflecting the estimated spatial distribution of the MWCNT interlayers in the nanocomposite. If we virtually remove the polymer from the nanocomposite, the highly segregated structure of MWCNTs would roughly resemble a foam structure with bubbles enveloping virtually removed polymer particles. However, the real observed structure is more complex than foam. Since the volume fraction of MWCNTs was low (~1 vol.%) and some surfaces of polymer particles remained uncovered by them (Figure 1c), the segregated structure was not completely interconnected. Furthermore, artificial features of image processing were also present, which typically appear when thresholding of images is performed at different threshold values. To fix the latter, series of 3D dilution and erosion operations were performed, allowing us to obtain the approximated 3D structure made of densely packed MWCNTs in the nanocomposite’s volume. Using the MeshLab software package, the obtained three-dimensional surface (Figure 4a) was simplified and corrected (Figure 4b), which later allowed importing it successfully into the Abaqus software package for further modeling using FEM.

A parallelepiped with dimensions 20 μm × 25 μm × 29 μm was created in Abaqus, to which the mechanical properties of UHMWPE were imparted. This parallelepiped was meshed uniformly with a cubic element edge length of ~0.5 microns. Subsequently, this matrix parallelepiped served as the “host” for the embedded element method, where the “embedded” elements were the elements of the segregated structure (Figure 4c,d).

The mesh for the filler structure was created using two-dimensional triangular Shell elements. The embedded element method implementation made it possible to significantly reduce the total number of elements for FE modeling (to ~2.5 × 10^5^ compared with the estimated > 2 × 10^6^ for the case of continuous meshing).

The considered 3D segregated structure of the nanofiller in the nanocomposite was composed of a spatial distribution of two-dimensional thin MWCNTs layers, differently oriented, enveloping polymer particles. When the nanocomposite was loaded by uniaxial deformation, MWCNTs layers were oriented at different angles relative to the axis of the applied loading, and their stress–strain state was determined via the combination of normal and shear loadings. Thus, to predict the piezoresistivity of the nanocomposite, it was necessary to involve multiscale modelling, where, at the nanoscale, the piezoresistive response of a 2D MWCNT layer was found at different angles of applied loading. Implementing then the obtained knowledge for the 3D structure of layers at the microscale, the macroscale piezoresistive response could be found.

A thin layer of MWCNTs was deformed uniaxially at different angles to its plane (Figure 5), thereby providing a combination of normal and shear forces in the layer. This corresponds to the concept of a three-dimensional structure of filler interlayers observed in the UHMWPE/MWCNT nanocomposite, in which, with uniaxial deformation, elementary interlayer segments were located at different angles to the axis of deformation of the nanocomposite. Uniaxial tension was applied in *Y*–*Z* plane of the layer at varying angles measured from the plane of the layer such that a 0° angle corresponded to the case when the axis of loading coincided with the *Y*-axis (in-plane loading) and a 90° angle corresponded to the case when the axis of loading coincided with the *Z*-axis (out-of-plane loading).

It was found numerically that relative electrical conductivity (electrical conductivity in the deformed state to the electrical conductivity in the undeformed state) of the layer always decreased linearly with the applied strain for all considered loading cases, $\frac{\mathrm{Electrical}\;\mathrm{conductivity}\;\mathrm{in}\;\mathrm{the}\;\mathrm{deformed}\;\mathrm{state}}{\mathrm{Electrical}\;\mathrm{conductivity}\;\mathrm{in}\;\mathrm{the}\;\mathrm{undeformed}\;\mathrm{state}}=\left(\mathrm{Slope}\left[{\%}^{-1}\right]\cdot \mathrm{strain}\left[\%\right]\right)$, and the results could be fully described by reporting the value of the slope. Obtained dependencies for the slopes of relative conductance versus angle of the applied loading are plotted in Figure 6a,b, with *X*-, *Y-*, and *Z*-directions denoting the anisotropic response of conductivity in these directions. Curves for the layer’s conductivity in *X*- and *Y*-directions (Figure 6a) were approximated with linear functions, the parameters of which were used in future analysis. The difference in conductivity in *X*-, *Y-*, and *Z*-directions (Figure 6c), as well as no notable trend for the slope values in the *Z*-direction (Figure 6b), allowed considering both the magnitude of the *Z* component of conductivity and its dependence on the loading direction as negligible.

At the next step, the obtained linear dependences for the conductivity changes in layers densely populated with MWCNTs were used for the homogenized elements of the segregated MWCNT structure to determine the segregated network response to deformation depending on the element’s orientation relative to the deformation direction. Due to the ability of linking the individual orientation to the properties of each 2D shell element (Figure 7a), the meshing was carried out using 2D triangular shell elements. It was possible to implicitly set the dependence of the anisotropic conductivity of each shell element on its orientation and the degree of uniaxial deformation of the investigated nanocomposite using subroutine UMAT in the FEA Abaqus software. Because, in the course of creation of the segregated structure surfaces as formed with shell elements, the structure was effectively doubled due to the initial thickness increase of the layer during the dilution–erosion operations (Figure 4a), the thickness of the shell elements was set as 50 nm. As conductivity of the matrix, a value of 10^−15^ S/m was used, while, for the MWCNT layer elements, a value of 10^5^ S/m was chosen. The precise values of the conductivity were not of the concern, since the contrast between conductivity of the matrix and densely packed MWCNTs was very high. As mechanical properties, values of 500 MPa and 5 GPa were chosen as elastic moduli for the matrix and MWCNT layers, respectively, and a Poisson coefficient of 0.46 was used for both matrix and MWCNT layers. These values were used because it was assumed that the filler did not influence mechanical properties of the matrix significantly since it was not integrated into the matrix, and it simply followed the matrix movements due to the friction forces. Unit potential difference was applied to the volume in the deformation direction, while the total current value passing through the top surface was measured for values of elongation in the range 0% to 5% with a 0.5% step (Figure 7b).

Comparison of the experimentally obtained results for UHMWPE + 2 wt.% MWCNT nanocomposite samples and the results of numerical simulations for the segregated structure are presented in Figure 8. The relative conductance vs. elongation dependency, obtained experimentally for the UHMWPE + 2 wt.% MWCNT nanocomposite, was close to the linear, with a slope of $\left(-6.6\pm 0.2\right)10^{-3}{\%}^{-1}$ being slightly lower than the slope of $\left(-5.21\pm 0.01\right)10^{-3}{\%}^{-1}$ from the numerical simulations of the segregated structure (the difference was in the scatter range of the experimental result).

Both discussed slopes, experimental and numerical, for the segregated structure were higher than the slope of $\left(-7.2\pm 0.2\right)10^{3}{\%}^{-1}$ for the 2D thin layer [36] when its loading was applied in plane (angle 0° in Figure 6a). This is easily explained by the dependences from Figure 6a; since the segregated structure combines MWCNT interlayers at all angles, the combined response would have a slope higher than the lowest point from Figure 6a. 

The difference between the experimental and numerical results may be explained by the presence of pores at the interface between polymer granules, observed for experimentally investigated samples. Because of that, the conductive pathways in the material were destroyed easier, and the whole nanocomposite sample had low mechanical properties, such as low elongation at break (~6%).

Taking into account the results, obtained numerically and experimentally for the UHMWPE + MWCNT nanocomposites (Figure 8), and the results, obtained for the thin layer of MWCNTs (Figure 6a) [36], it can be said that the assumptions and approaches made for the numerical simulations can successfully act as a foundation of a working model, capable of predicting the deformational behavior of nanocomposites with a segregated structure made of an electroconductive nanosized filler, such as MWCNTs. By correctly evaluating the impact of all modeling steps on the piezoresistive response of the nanocomposite, it might be possible to find ways to determine the experimental flaws, which can be dealt with or simply accounted for in the future iterations of the model. 

Comparing our results to previous works, we can state that comparable accuracy of the numerical predictions was achieved but our approach of finite element simulations on the basis of embedded elements significantly reduces requirements for computer resources because this technique does not require detailed meshing on the interface between two composite phases. This allowed us to model the nanostructure in more detail combining calculations of different scales.

The numerical models based on the proposed approaches can find various applications where computationally effective simulations of systems with densely packed filler particles (with volume fractions well above the percolation threshold), such as MWCNTs, are required, and where the filler is distributed by thin layers inside a nonconductive phase. An example of such systems is the nanocomposite based on solid-state processed UHMWPE reactor powder, which is mechanically premixed with MWCNTs to obtain the extreme state of filler segregation. These types of materials can find many interesting applications in the future due to their ability to be further processed in a solid state in order to achieve a highly oriented state and high mechanical characteristics while still maintaining high electrical conductivity.

## Figures and Tables

**Figure 1 nanomaterials-11-00162-f001:**
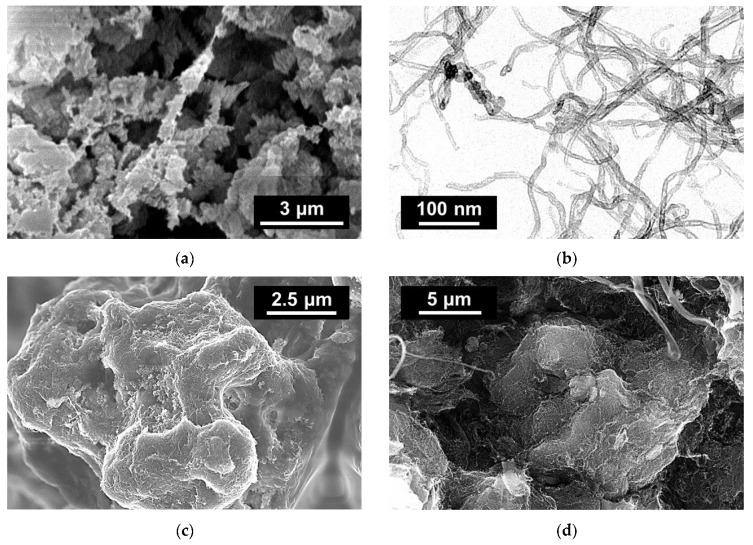
Results of scanning (**a**,**c**,**d**) and transmission (**b**) electron microscopy for (**a**) the ultrahigh-molecular-weight polyethylene (UHMWPE) reactor powder surface, (**b**) multiwalled carbon nanotubes (MWCNTs), (**c**) the UHMWPE powder surface covered with 2 wt.% MWCNTs, and (**d**) the UHMWPE + 2 wt.% MWCNT nanocomposite fracture surface.

**Figure 2 nanomaterials-11-00162-f002:**
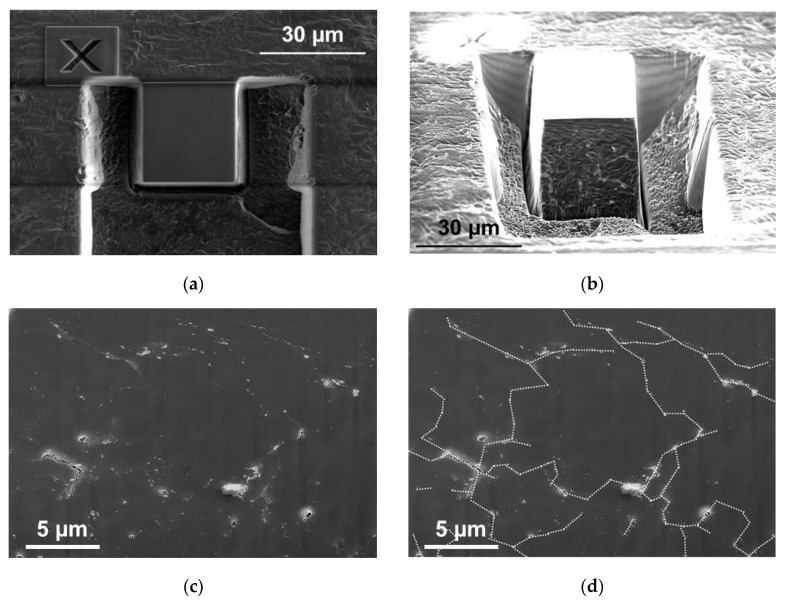
(**a**,**b**) Focused ion beam scanning electron microscopy images of the investigated volume of the UHMWPE + 2 wt.% MWCNT nanocomposite; (**c**,**d**)one of the resulting slice images: (**c**) as it was and (**d**) with distribution of areas highly populated with MWCNTs reconstructed by processing images in sequence (dotted white lines).

**Figure 3 nanomaterials-11-00162-f003:**
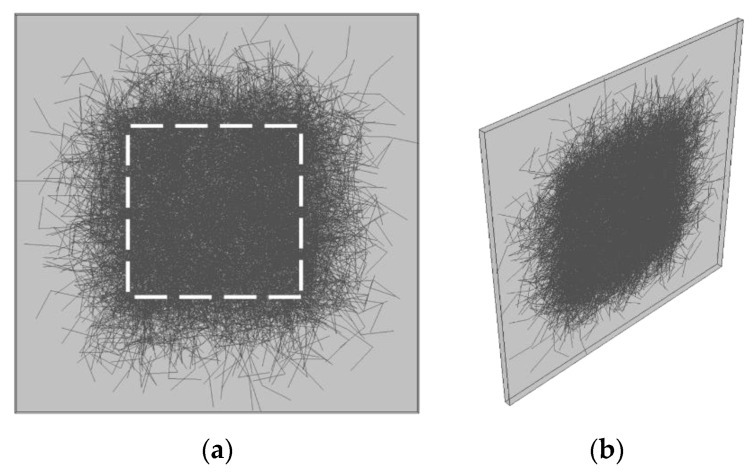
(**a**) Volume element constructed for deformation tests and conductance calculations with imitated periodic boundary conditions, where the white dashed square contours the investigated sub-volume (“inner” volume), to which the deformation was applied and the conductance of which was calculated later. (**b**) Side view of the volume element demonstrating its thin layer structure confined in between two polymer particles.

**Figure 4 nanomaterials-11-00162-f004:**
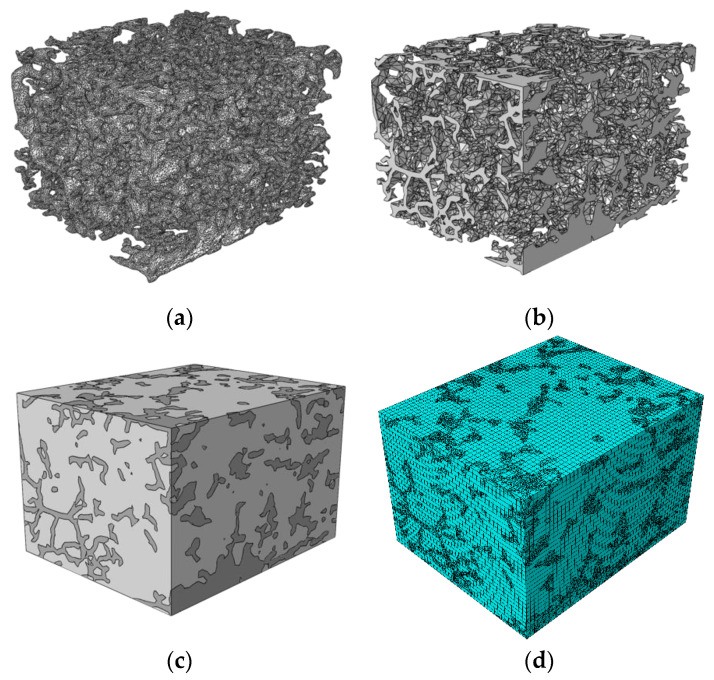
Three-dimensional (3D) triangular shell element meshes of the reconstructed structure made of densely packed MWCNTs in the UHMWPE + 2 wt.% MWCNT nanocomposite (**a**) before and (**b**) after the simplification and correction; (**c**) geometry reflecting distribution of densely packed MWCNTs (dark gray) in UHMWPE matrix (light gray) reconstructed for UHMWPE + 2 wt.% MWCNT nanocomposite sample; (**d**) result of separate meshing of the matrix parallelepiped volume and MWCNT formed surface followed by embedding of the filler elements into the matrix element mesh. At a closer look, the two meshes do not coincide.

**Figure 5 nanomaterials-11-00162-f005:**
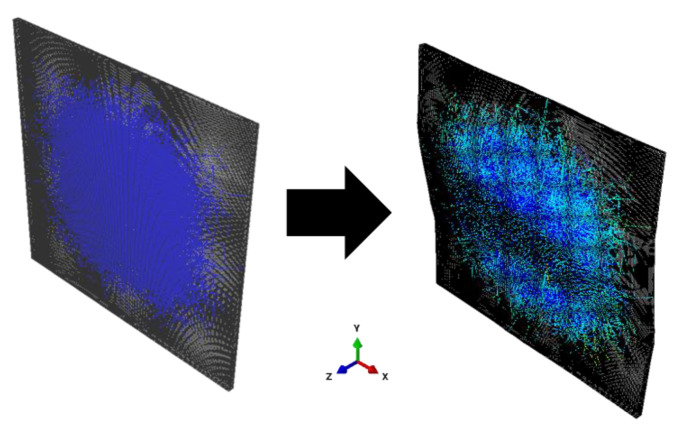
Scheme of deformation application under certain angles to the plane of the layer for the investigated thin volume containing MWCNTs (additional colors on the right image illustrate the tensile stress distribution in MWCNT elements).

**Figure 6 nanomaterials-11-00162-f006:**
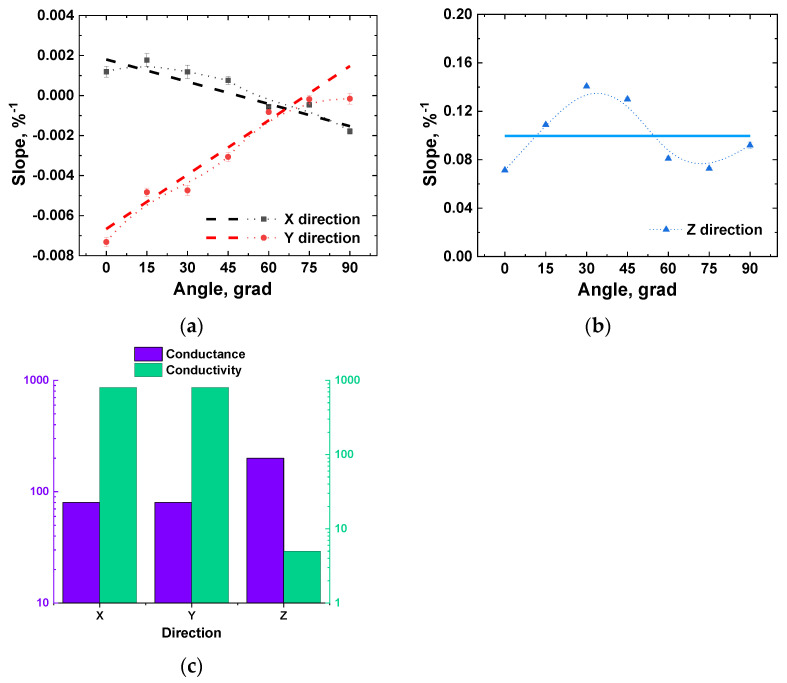
Values of slopes of the conductivity dependencies from elongation for all directions (*X* and *Y* (**a**), and *Z* (**b**)) of the thin layer of MWCNTs versus the angle between the deformation direction and *Y* direction. (**c**) Anisotropic values of layer conductance (in units of MWCNT joint conductance) and conductivity (previous value of conductance transformed into conductivity by the layer dimensions) in *X*-, *Y*-, and *Z*-directions for the loading angle = 0. Coordinate directions are from Figure 5.

**Figure 7 nanomaterials-11-00162-f007:**
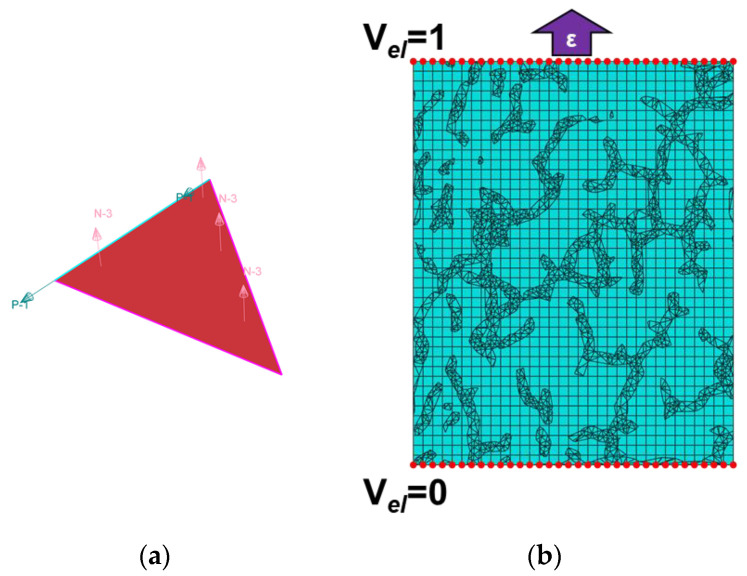
(**a**) Illustration of a shell element with set orientation; (**b**) scheme of deformation application with simultaneous conductance measurement for the investigated volume of nanocomposite with segregated structure used in the model.

**Figure 8 nanomaterials-11-00162-f008:**
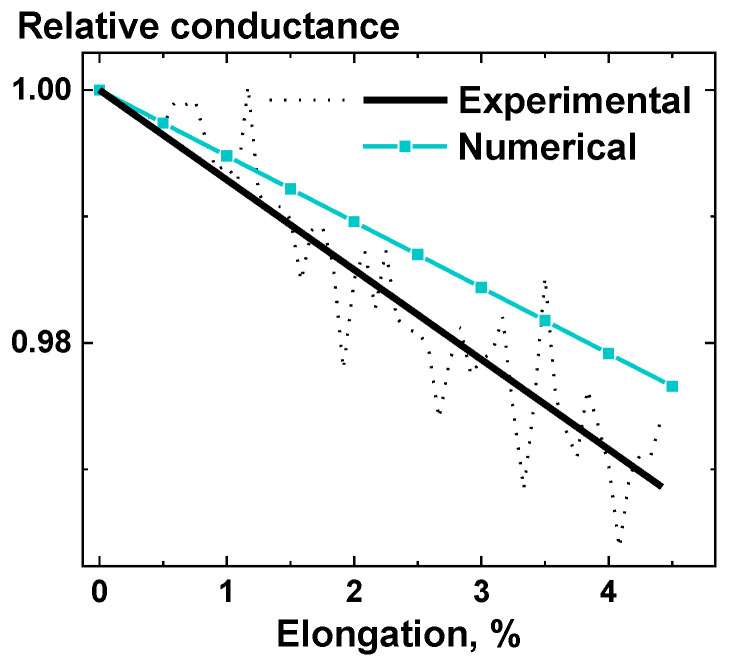
Experimental results for relative conductance response of UHMWPE + 2 wt.% MWCNT nanocomposites to deformation and numerical results, obtained for segregated structure reconstructed with tomography. For experimental results, the raw data averaged for three experiments (black dashed line) and the linear fit (black solid line) are presented.

**Table 1 nanomaterials-11-00162-t001:** Specification of the Nanocyl™ NC 7000 MWCNTs [38].

Parameter	Method of Testing	Value
Specific area, m^2^/g	**Brunauer–Emmett–Teller**	250–300
Purity, %	Thermogravimetric analysis	90
Average diameter, nm	Scanning electron microscopy	9.5
Average length, nm	Scanning electron microscopy	1500
Metal content, %	Inductively coupled plasma mass spectrometry	<1
Resistivity, Ω∙cm	Powder conductivity measurements	10^−4^

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
