# Peer review of "Multiscale Numerical Modeling for Prediction of Piezoresistive Effect for Polymer Composites with a Highly Segregated Structure"

_nanomaterials, 2021, doi:10.3390/nano11010162_

Round 1

Reviewer 1 Report

The authors performed a numerical analysis of the piezoresistive effect on UHMWPE/MWCNTs and compared their results with experimental studies. The authors should apply my comments to proceed with the publication of the paper.

  • How the authors defined "highly segregated" structuures. It is better to use just a segregated structure.
  • Authors should work on the level of English in the manuscript because in some parts it is extremely difficult to follow.
  • Authors use distribution and dispersion as two key factors for predictable electrical and mechanical properties of polymer nanocomposite. It is highly recommended to show dispersion and distribution of the filler graphically for a better understanding of the readers. Usually, 4 figures were depicted to show good/bad dispersion with good/bad distribution. Do the authors think if a very uniform “distribution” is achieved is it good for electrical properties or not? Does that make it difficult for reaching percolation?
  • For CNT in general, a very low percolation threshold is reported for different systems. Unlike 2D materials, CNTs can disperse well in polymers with melt mixing and this facilitates to reach percolation threshold at low CNT contents.
  • Please rewrite “It was demonstrated earlier that creation within a polymer material a structure made of filler particles, characterized by denser filler packing compared 59 to the rest composite volume (so-called segregated structure), is a good way to significantly reduce 60 percolation threshold value for the resulting composite material” since it is difficult to understand.
  • Please used compression instead of compactification which has another meaning
  • The authors should report how much polymer, MWCNT, and hexane were used for composite preparation.
  • To verify the segregated structure, the authors should also add the optical microscopy image of microtome samples. There is no visual evidence of segregated structure was introduced. The contrast in Figure 2-c is very poor and the difference between CNT and polymer is very difficult.
  • Does Figure 3 represent the true nature of the polymer nanocomposite? Because it is agglomerated with very bad distribution and dispersion and also very high-volume content which is different from their prepared samples. And how did authors account for “segregated structure” in this model?
  • It is not clear how the authors concluded by just comparing the percolation thresholds that their system is segregated. The percolation threshold can be lowered by enhancing the dispersion as well.
  • Remove the “division” symbol from line 251.
  • What is “relative conductance”?

Author Response

Review 1

Dear Reviewer,

Thank you for your comments and corrections! Please find below our answers. All comments have been answered with necessary corrections implemented and are highlighted in green.

-How the authors defined "highly segregated" structures. It is better to use just a segregated structure.

---Two processing ways for the UHMWPE have been considered in our study: hot sintering (above the melting point) and solid state processing. In the second case, the segregation of the filler particles is extreme while in the first case due to melting some filler particles may be dispersed wider. Therefore, we reserve the term “highly segregated structure” to the structure formed in the case of solid processing. And additional explanation has been added to the text. (lines 87-96)

-Authors should work on the level of English in the manuscript because in some parts it is extremely difficult to follow.

---Extensive English language corrections have been implemented throughout the text.

-Authors use distribution and dispersion as two key factors for predictable electrical and mechanical properties of polymer nanocomposite. It is highly recommended to show dispersion and distribution of the filler graphically for a better understanding of the readers. Usually, 4 figures were depicted to show good/bad dispersion with good/bad distribution. Do the authors think if a very uniform “distribution” is achieved is it good for electrical properties or not? Does that make it difficult for reaching percolation?

---In our study, we investigate only the highly segregated structures in the result of solid state processing. In this case, a “dry” CNT powder envelopes polymer powder granules, pressed then together. In this case, we cannot talk about dispersion of CNTs in the matrix because no such dispersion is present for processing below the melting point. Present are only 3D distribution of segregated structures in the nanocomposite and distribution of “dry” CNTs on powder surfaces. Thereby, we used the term dispersion only in the introduction for the reference to the hot processed nanocomposites and do not use it in the main body of the manuscript. The 3D distribution of segregated structure is presented in Fig. 4 and the distribution of CNTs on polymer granule surfaces is presented in Fig. 1. Additional clarification has been added to the text. (lines 79-80, 87-96).

-For CNT in general, a very low percolation threshold is reported for different systems. Unlike 2D materials, CNTs can disperse well in polymers with melt mixing and this facilitates to reach percolation threshold at low CNT contents.

---There was a misunderstanding in the text. We did not want to say that uniform dispersion of CNTs has high percolation threshold, we wanted to say that when uniform dispersion is aimed but the actual dispersion is bad, then we have high percolation threshold. Corrected. (lines 56-58)

-Please rewrite “It was demonstrated earlier that creation within a polymer material a structure made of filler particles, characterized by denser filler packing compared 59 to the rest composite volume (so-called segregated structure), is a good way to significantly reduce 60 percolation threshold value for the resulting composite material” since it is difficult to understand.

---Rewritten. (lines 61-72)

-Please used compression instead of compactification which has another meaning

---Corrected. (lines 86, 162, 169-172)

-The authors should report how much polymer, MWCNT, and hexane were used for composite preparation.

---Reported. (lines 153-155)

-To verify the segregated structure, the authors should also add the optical microscopy image of microtome samples. There is no visual evidence of segregated structure was introduced. The contrast in Figure 2-c is very poor and the difference between CNT and polymer is very difficult.

---We understand that it is hard to depict a segregated structure using microscopy at such low MWCNTs content due to significant differences between sizes of the polymer granules and the thickness of the MWCNTs layer. Thus, a large number of electron microscopy images were studied, thus giving an additional evidence of the presence of the highly segregated structure. Especially it was seen when the slices, obtained using focused ion beam etching, are studied consecutively as a large set. Unfortunately, it is not possible to present a sufficient number of microscopy images in the manuscript to remove all the doubts due to the limitation on the manuscript volume and to prevent confusion of the reader, who will have to compare all the images. That’s why in Fig. 2d a dashed line is given, demonstrating reconstructed segregated structure, taking into account distribution observed at the multiple slices in front and in the back of the current slice. Due to the effect of gel formation when the mixture of UHMWPE + MWCNTs is processed in the hexane using ultrasound at very high contents of MWCNTs, it was not possible to obtain the corresponding samples with uniform MWCNTs distribution. At the same time, at low contents of MWCNTs the optical microscopy is not very effective. The segregated structure of the composites, based on UHMWPE, was also obtained by us for other types of fillers, such as carbon black and graphite nanoplatelets, at high contents of the filler. Microscopy data demonstrate a clear segregation in these cases. Unfortunately, we are yet to finish modeling and experimental studies of these systems, and introduction of the preliminary results in the article seems to be inappropriate. (lines 192-199)

-Does Figure 3 represent the true nature of the polymer nanocomposite? Because it is agglomerated with very bad distribution and dispersion and also very high-volume content which is different from their prepared samples. And how did authors account for “segregated structure” in this model?

---In the result of solid state processing (below polymer melting point) there is no dispersion of CNTs in the matrix, only CNTs covering polymer powder surfaces. Relevant discussions have been added to the text and are described in answers to previous comments. Yes, this model does represent the true nature of high segregation.

-It is not clear how the authors concluded by just comparing the percolation thresholds that their system is segregated. The percolation threshold can be lowered by enhancing the dispersion as well.

---Again, this is confirmed by Fig 1d and by the nature of the manufacturing process – no melting is involved. (lines 87-96, 165-169)

-Remove the “division” symbol from line 251.

---Removed. (line 296)

-What is “relative conductance”?

---The definition is given in section 2.2.2: “the relative electrical conductivity — the ratio of the value of electrical conductivity in the deformed state to its value in the undeformed sample” (lines 187-188)

Best regards,

The authors

Reviewer 2 Report

This manuscript reports a numeric simulation method to predict the piezoresistive effect of CNT/HMWPE composites with a segregated structure. While the model seems to predict the experimental results in a relatively accurate way, I do not see the particular advantages of such a model in comparison with other existing models in the literature. Therefore, I would suggest a major revision to highlight the benefits of the proposed model. Specific comments follow:

  1. Figure 2c: the percolated structure of CNTs cannot be seen clearly because of the dash lines. I would suggest removing the dash lines.
  2. Figure 6a: what is the slope and angle? Detailed explanation should be provided for the two parameters and justify why there is a linear relationship between the two.
  3. Figure 6c: the units for conductance and conductivity are missing. Explain why the conductance/conductivity is the same in x- and y-directions.
  4. Figure 8: the authors should compare the results of their model with others in the literature and clearly state the benefit of their model.

Author Response

Review 2

Dear Reviewer,

Thank you for your comments and corrections! Please find below our answers. All comments have been answered with necessary corrections implemented and are highlighted in yellow.

This manuscript reports a numeric simulation method to predict the piezoresistive effect of CNT/HMWPE composites with a segregated structure. While the model seems to predict the experimental results in a relatively accurate way, I do not see the particular advantages of such a model in comparison with other existing models in the literature. Therefore, I would suggest a major revision to highlight the benefits of the proposed model. Specific comments follow:

  1. Figure 2c: the percolated structure of CNTs cannot be seen clearly because of the dash lines. I would suggest removing the dash lines.

The requested figure is presented below and is included in the manuscript (fig. 2c)

  1. Figure 6a: what is the slope and angle? Detailed explanation should be provided for the two parameters and justify why there is a linear relationship between the two.

A detailed explanation has been added to the text. (lines 357-375)

  1. Figure 6c: the units for conductance and conductivity are missing. Explain why the conductance/conductivity is the same in x- and y-directions.

Clarification is added to the figure capture. The conductance/conductivity is the same in x- and y-directions from figure 5 because both these directions are in the plane of the layer and their response is identical.

  1. Figure 8: the authors should compare the results of their model with others in the literature and clearly state the benefit of their model.

A general statement has been added to the text. (lines 435-439) To the best knowledge of the authors, no other multi-scale numerical modeling of the composites with highly segregated structure was performed for the UHMWPE+MWCNT nanocomposite. First, because such composite systems, characterized by the extreme segregation, obtained using processable in a solid-state UHMWPE, are novel and were first obtained and studied by the authors. Second, due to limitation of FEM, it is not possible do perform numerical simulation of very densely packed filler particles, especially 1D (as it was done in our work), without implementing embedded element method and creating a special volume element that complies with the method specifics. This was also done by the authors for the first time. All these facts are mentioned in the text and literature review of this manuscript.

Best regards,

The authors

Reviewer 3 Report

please see separate file attached

Author Response

Review 3

Dear Reviewer,

Thank you for your comments and corrections! Please find below our answers. All comments have been answered with necessary corrections implemented and are highlighted in purple.

Lebedev, Ozerin and Abaimov present a combined experimental and numerical description of a conducting polymer blend, where the conducting component is based on multi-walled carbon nanotubes. As a theorist, I can mainly comment on the simulation and on the numerical results, but not on the experiments performed therein. From my perspective, the research presented in the manuscript is novel and original and should be of interest to polymer scientists working in the field of organic electronics and nano-structured systems. The numerical methods are sound and up to date. I have enjoyed reading the paper, which is well written, concise and provides a very good introductory section appropriate for non-experts. I was first puzzled by the fact that no proper periodic (or minimum image) boundary conditions have been used. However, this is due to the methodology applied in the paper, and rationalized in detail by the authors.

I have a few minor points that I would like the authors to respond to prior to a publication of the results:

-Second page, last paragraph, around line 92: the two-mesh approach is valid only if the two components strongly differ in their bulk conductivities. Can the authors provide typical values for the pure components here (extrapolated for the MWCNTs, if applicable) ?

---To the best of our knowledge, the general practice of the two-mesh procedure in literature does not require high contrast in properties because its idea is just to identify and connect closely located nodes between two mesh networks without any regard to conductivity – the error comes only from tolerance with which nodes are associated which is typically one element width. Utilized in our calculations conductivities are listed in the text: “As conductivity of the matrix, a value of 10^-15 S/m was used, while for the MWCNTs layer elements value of 10^5 S/m was chosen.” (lines 392-393)

-Title: I would not really call the parametrization of a model on a small system and the consecutive simulation in bulk 'multi-scale'. This should, however, be left to the discretion of the editor.

---Three scales are involved: nano-scale for the part of 2D layer simulations, micro-scale for the 3D structure of layers, and macro-scale of the nanocomposite. In the literature, this is typically called multi-scale. (lines 128-131)

-Section 3, line 244 and following. I do not really understand the argument. Usually, segregation leads to an increase of the percolation threshold by removing narrow, conducting filaments between islands of conducting material. References should be given for the pc value of 0.4 vol % mentioned there. This seems extraordinarily small to me, I would expect a volume fraction (not percentage) of 0.17 for 3D continuum percolation (numerical results), or a mean-field value of 1/3.

---In our system, the segregation leads to the decrease in the percolation threshold. Comparing uniform distribution of CNTs with the foam-like structure formed around polymer granules, the latter needs less CNT material to create a percolating path, since only part of polymer is enhanced with CNTs leaving insides of granules additive-free. Experimental results for CNTs demonstrate percolation thresholds much lower than 0.5wt.%. The exact value cannot be determined experimentally due to agglomeration, but values as low as 0.1wt.% and probably 0.05wt.% can be observed. These correspond to volume fractions around 0.03-0.05%. The cited theoretical values 17% and 33.3% are probably not for particles with the aspect ratios like of CNTs that is of the order of 5 micrometer / 50 nanometer = 100. (lines 66-72)

-Equation 1. To make a connection to mainstream physics, it would be helpful to refer to this as Kirchhoff’s law.

---Corrected. (lines 272, 282)

-Figure 6. Please add units to the axis or to the figure caption.

---Units for Figure 6c are added to the caption. Units for figures 6a and 6b are %^-1 as inversed units of strain measured in %. (lines 368, 379, 380)

-In general, it would be very helpful to have a table giving an overview over the simulation details, such as composition, number of realizations, system size etc.

---The manuscript is written in the style maximally avoiding units. For example, we consider relative conductivities as conductivities of deformed state normalized by conductivities of the undeformed state. The results of the 2D calculations can be read from Figures 6a,b while Figure 8 provides results for the 3D segregated structure under loading. We afraid that adding a table combining parameters for mechanical 3D, electrical 3D, mechanical 2D, electrical 2D simulations will only confuse a reader.

-There is something missing in line 164, ‘...that in numerical simulations of [???] filled with...’

---Corrected. (lines 205-208)

Best regards,

The authors

Round 2

Reviewer 1 Report

The authors extensively their report and respond well to my concerns. I advise the editor to accept this paper for the publication.

Reviewer 2 Report

The revision has addressed the comments raised previously. The manuscript can be accepted in the current form.